# Characterization of Functional Microorganisms in Representative Traditional Fermented Dongcai from Different Regions of China

**DOI:** 10.3390/foods12091753

**Published:** 2023-04-23

**Authors:** Yanbing Jiang, Hao Fu, Meng Li, Changtao Wang

**Affiliations:** 1Beijing Key Laboratory of Plant Resource Research and Development, College of Chemistry and Materials Engineering, Beijing Technology and Business University, Beijing 100040, China; 2130042087@st.btbu.edu.cn (Y.J.); 18811359600@163.com (H.F.); wangct@th.btbu.edu.cn (C.W.); 2Institute of Cosmetic Regulatory Science, Beijing Technology and Business University, Beijing 100040, China

**Keywords:** Dongcai, different regions of China, high-throughput sequencing (HTS), microbial diversity

## Abstract

Dongcai is loved for its delicious flavor and nutritional value. The microorganisms in Dongcai play a vital role in their flavor, quality, and safety, and the microbial communities of Dongcai vary greatly from region to region. However, it remains unknown what the predominant microorganisms are in different traditional Dongcai and how they affect its flavor. The objective of this study is to explore the microbial diversity of traditional fermented Dongcai in three representative Chinese regions (Tianjin, Sichuan, and Guangzhou) and further assess their microbial functions. The microbial diversity of fermented Dongcai in Guangdong has the lowest diversity compared to fermented Dongcai in Sichuan, which has the highest. The distribution of the main genera of fermented Dongcai varies from region to region, but *Carnimonas*, *Staphylococcus*, *Pseudomonas*, *Sphingomonas*, *Burkholderia-Caballeronia-Paraburkholderia,* and *Rhodococcus* are the dominant genera in common. In addition, halophilic bacteria (HAB, i.e., *Halomonas Bacillus*, *Virgibacillus*, etc.) and lactic acid bacteria (LAB, i.e., *Weissella* and *Lactobacillus*) are also highly abundant. Of these, *Burkholderia-Caballeronia-Paraburkholderia*, *Rhodococcus*, *Sphingomonas*, *Ralstonia,* and *Chromohalobacter* are dominant in the Sichuan samples. In the Tianjin samples, *Lactobacillus*, *Weissella*, *Virgibacillus*, *Enterobacter*, *Klebsiella,* and *Pseudomonas* are the most abundant. Predictions of microbial metabolic function reveal that carbohydrates, amino acids, polyketides, lipids, and other secondary metabolites are abundantly available for biosynthesis. In addition, the different flavors of the three types of Dongcai may be due to the fact that the abundance of HAB and LAB shows a significant positive correlation with the amounts of important metabolites (e.g., salt, acid, amino nitrogen, and sugar). These results contribute to our understanding of the link between the distinctive flavors of different types of Dongcai and the microorganisms they contain and will also provide a reference for the relationship between microbial communities and flavor substances in semi-fermented pickles.

## 1. Introduction

Dongcai is a semi-dried pickled vegetable, one of the famous Chinese specialties, which is made from whole mustard (*Brassica juncea*) or Chinese cabbage (*Brassica rapa*) without roots. Dongcai can be classified according to its place of origin, such as Sichuan Dongcai, Guangzhou (Chaoshan flavor) Dongcai, and Tianjin Dongcai. According to historical records, Nanchong Dongcai in Sichuan has been around for more than 1700 years and has become famous overseas since the Jiaqing Period of the Qing Dynasty. Sichuan Dongcai is made from mustard, while Guangzhou (Chaoshan flavor) Dongcai and Tianjin Dongcai are made from Chinese cabbage. Dongcai is a traditional condiment with a high sodium chloride content. Sichuan Dongcai is loved by the Chinese for its unique flavor, which develops during its long curing and fermentation process. However, it is characterized by a long curing cycle, low production, and unstable product quality. Tianjin Dongcai is called “meat Dongcai” if about 20% of garlic is added and “vegetarian Dongcai” if no garlic is added [1,2,3].

Fermentation is a widely used technique for preserving food, improving nutritional value, and extending shelf life, and fermented vegetables are very popular and traditional in Asian countries [4]. Microbial fermentation produces substances, spice substances, and raw material hydrolysis substances that react chemically, and the resulting substances work together to give Dongcai its special color, aroma, and flavor. Modern scientific research has confirmed that fermented foods have a variety of nutritional and health functions, and mustard is rich in amino acids after pickling, with the highest content of glutamic acid and aspartic acid [5,6,7]. Many studies have been carried out on the diversity of bacterial communities in kimchi and sauerkraut, but none have been done on the diversity of bacterial communities in Chinese Dongcai. Therefore, it is important to study the composition and diversity of bacterial communities in Dongcai, which can provide references for the development and use of bacterial resources and the production of artificial inoculum ferments.

Previously, only conventional microbial detection and techniques that depended on biomolecular culture have been used to research Dongcai microorganisms. Recently, however, high-throughput sequencing (HTS) methods have been progressively developed, leading to their widespread use for monitoring microbial communities in the fermentation of various foods and beverages, as well as vegetables [8,9]. The networks of connections between microbial functional groups and the development of fermented flavor in Dongcai may be significant. Therefore, the study of the functional genomes of microbial populations by HTS technology can help to reveal the core functional microbiota in Chinese Dongcai and its relationship with the flavor of Dongcai. 

In this study, the HTS approach was used to analyze the microbiological diversity of Dongcai from three different locations in China. The results of this study improve the understanding of the microbial diversity of Dongcai and provide important guidance for the future use of beneficial microbial resources.

## 2. Materials and Methods

### 2.1. Dongcai Sample Preparation

Three different varieties of Dongcai, all of which are natural fermentation products, were purchased from Tianjin (TJ), Sichuan (SC), and Guangzhou (GZ) in China. The three samples of SC were named: DFY11, DTG12, and DZY13; the three samples of GZ were named: DBT22, DLY22, and DXJ23; the three samples of TJ were named: DQY32, DFN31, and DCC33. Three parallels were made for each sample, and there were 27 samples in total (Table 1). Before extracting the DNA, Dongcai samples were collected in sterile test tubes and kept at −80 °C.

### 2.2. Determination of the Content of Salt, Acid, Amino Nitrogen, and Sugar

Analysis of the total acid content was done in accordance with GB/T 12456-2021 [10], the amino acid content in accordance with GB 5009.124-2016 [11], the sugar content in accordance with GB 5009.8-2016 [12], and the salt content in accordance with GB 5009.42-2016 [13].

### 2.3. DNA Extraction and PCR Amplification

Nine samples from three different regions, each with three duplicates, had their total DNA extracted using DNA extraction kits in strict accordance with the manufacturer’s instructions. A NanoDrop 2000 UV-Vis spectrophotometer (Thermoscientific, Waltham, MA, USA) was then used to analyze the DNA concentrations. Using the thermal cycling PCR technique, the V3-V4 hypervariable region of the bacterial 16S rRNA gene was amplified (Gene Amp 9700, ABI, Thermo Fisher Scientific Co., Ltd., Waltham, MA, USA). With three duplicates of each sample, all tests were conducted in accordance with formal experimental guidelines. PCR products were recovered by gel cutting with the AxyPrepDNA Gel Recovery Kit (AXYGEN, Shanghai, China), eluted with Tris HCl, and identified by 2% agarose gel electrophoresis. PCR products from the same sample were mixed together and detected using this method.

### 2.4. MiSeq Sequencing

According to the common technique described by Majorbio Bio-pharm Technology Co., Ltd. (Shanghai, China), the purified and amplified fragments obtained were pooled on an Illumina MiSeq platform for equimolar and paired-end sequencing (2 × 300).

### 2.5. Data Analysis Process

The PE reads obtained from the MiSeq sequencing were first spliced according to the overlap relationships, while the sequence quality was quality controlled and filtered, and the samples were differentiated and then subjected to operational taxonomic unit (OTU) clustering analysis and taxonomic analysis of species. Based on the results of OTU clustering analysis, multiple diversity index analyses were performed, as well as the depth of sequencing based on taxonomic information, and the statistical analysis of community structure was carried out at all taxonomic levels. Based on these analyses, a series of in-depth statistical and visual analyses were carried out on the community composition and phylogenetic information of the samples, including multivariate analyses and difference significance tests.

### 2.6. Data Analysis

The free Majorbio I-Sanger Cloud Platform was used to conduct HTS analysis (www.I-Sanger.com, accessed on 16 March 2020). Both QIIME software (version 1.9.1; available at http://qiime.org, accessed on 16 March 2020) and Arch software (version 7.0; available at http://www.drive5.com, accessed on 16 March 2020) were used, as well as the Mothur program (version 1.30.2, available at mothur.org, accessed on 16 March 2020). The Observed, Chao1, and ACE indices were used to measure community diversity, while the Shannon and Simpson indices were used to measure community richness. PICRUSt (version 1.1.0, http://PICRUSt.github.io, accessed on 16 March 2020) was used to forecast the distribution of homologous gene clusters (COGs) based on 16S rRNA gene sequences. 

To assess their differences, an analysis of alpha diversity was conducted using the Wilcoxon test. Based on Bray–Curtis phase dissimilarity, beta diversity was evaluated and visualized using principal coordinate analysis (PCoA). To evaluate the differences between the two groups, an analysis of similarity (ANOSIM) was conducted. R was used to conduct the statistical analysis and mapping of PCoA and ANOSIM. Additionally, linear discriminant analysis (LDA) was used to perform biomarker analysis of effect sizes (LEfSe) in order to pinpoint the bacterial taxa most likely to result in taxon demarcation.

## 3. Results

### 3.1. Microbial Annotation and Evaluation

There were 27 Illumina Novaseq 6000 sequencing samples in total. A total of 1,074,228,666 16S rRNA raw bases and 671,906,243 good-quality 16S rRNA bases, with an average length of 377 bases, were acquired. The bases in the validated tag (Q20) of each sequence amplicon with a minimum base calling accuracy of 98% were all greater than 97%, indicating that the accuracy of the sequencing data was high and could be used for further research. The total number of OTUs for bacteria grouped by species was 1683, with 1064 for TJ, 940 for GZ, and 1200 for SC.

The Sobs sparse curve analysis revealed that the curve was not parallel to the x-axis (Figure 1A) but eventually began to flatten out, with over 99% coverage of bacteria. The results indicate that DZY13 had the highest species diversity, and DTG12 had the lowest.

To analyze the characteristics of microorganisms shared at the endemic level in different Dongcai, a Venn diagram analysis was carried out based on OTU abundance. A total of 3016 OTUs were discovered from nine samples, including 1213 from SC, 942 from GZ, and 1064 from TJ, as shown in Figure 1B. The total number of OTUs shared by the three Dongcai was 560, with 400 unique OTUs from SC, indicating a much lower microbial diversity than the Dongcai from the other two regions. 

Based on the observed Chao1 and ACE indices, significant differences (*p* < 0.05) in the abundance of the microbial communities in Dongcai were found between the TJ, SC, and GZ samples (Figure 2). This indicated that the TJ Dongcai community had the highest bacterial community abundance, with GZ the second lowest and SC the lowest. According to an analysis of the Shannon and Simpson indices of the samples from the three regions (*p* > 0.05), there was no significant difference in microbial community diversity between the SC and TJ samples, suggesting that the bacterial homogeneity and community coverage of the microbiota in the Dongcai from the two regions were similar and significantly higher than those of GZ Dongcai.

Figure 3A depicts the dynamics of the bacterial community at the phylum level in Dongcai from various regions, including the distribution of major bacterial groups, in which a total of 28 bacterial phyla were discovered. The top four phyla were Aspergillus, Thick-walled bacteria, Bacteroides, and Actinobacteria. At the genus level, 772 different bacterial genera were found. The top 10 genera were *Carnimonas*, *Bacillus*, *Weissella*, *Staphylococcus*, *Pseudomonas*, *Lactobacillus*, *Virgibacillus*, *Sphingomonas*, *Burkholderia-Caballeronia-Paraburkholderia,* and *Rhodococcus* (Figure 3B). The overall characteristics of the bacterial composition of Dongcai from the three regions were made up of these bacterial taxa.

Based on weighted UniFrac distances, PCoA (Figure 4A) revealed that the composition of the different species of bacteria in the bacterial communities explained 28.67% of the variation in community structure, followed by PCo2 (24.64%). Anasim tests revealed that the differences in microbial communities between groups were higher than those within groups (bacteria: Unweighted UniFrac = 0.5860, *p* = 0.001). These findings imply that classification by region is appropriate.

Variation in microbial communities in Dongcai from different regions was further validated by β-diversity analysis using non-metric multidimensional scaling (NMDS) (Figure 4B). The apparent separation between the SC, GZ, and TJ inoculum groups indicated that different microbial communities existed in different regions.

Using the LDA LEfSe approach, it was possible to identify biomarkers in the bacterial community composition that varied significantly between the SC, GZ, and TJ samples. The LEfSe program enables statistical analysis from the phylum level to the species level, as well as the examination of data from any branch of the microbial community. As shown in Figure 5, Actinobacteria was predominantly sampled from SC, while Bacteroides was predominantly sampled from TJ, which is consistent with previous microbiome analyses. In addition, 36 bacterial branches showed significant differences with an LDA threshold of 4.0. Five genera in the SC samples (*Burkholderia-Caballeronia-Paraburkholderia*, *Rhodococcus*, *Sphingomonas*, *Ralstonia,* and *Chromohalobacter*) were significantly enriched, one genus (*Carnimonas*) dominated the GZ samples, and six genera (*Lactobacillus*, *Weissella*, *Virgibacillus*, *Enterobacter, Klebsiella,* and *Pseudomonas*) were enriched in the TJ samples, resulting in the differences between the three regions.

### 3.2. Correlation Analysis of Microbial Communities and Biochemical Indicators of Dongcai

The flavor, quality, and stability of fermented Dongcai are influenced by a combination of the four ingredients: salt, acid, amino nitrogen, and sugar. Fermented Dongcai from the three regions differed in these four aspects (Table 2). In addition, the correlation between microorganisms and environmental variables was assessed at the level of the genus in the samples, as visualized by the heat map (Figure 6). The highest levels of salt were found in the TJ samples, probably due to the high amount of salt added during processing. We found that Weissella, Pantoea, Klebsiella, Lactobacillus, Enterobacter, Idiomarina, and Pseudomonas showed a significant positive correlation with salt, while Carmonas showed a negative correlation. Samples from SC had the highest levels of acid and amino nitrogen compared to the other two sites. Further analysis revealed that acid content made the fermentation system’s pH lower, which dramatically lowered the development of Vibro, Idiomarina, and Halomonas while significantly boosting the growth of Carmonas. In addition, Staphylococcus and Corynebacterium had higher abundance in the samples with high amino nitrogen content. The highest sugar content was found in the GZ samples, which greatly promoted the growth of Sphingomonas, Burkholderia-Caballeronia-Paraburkholderia, and Rhodococcus. 

### 3.3. Functional Prediction of Microbial Community Structure

To compare the functional potential of microorganisms, predictions of OTU-based functional abundance were made using PICRUSt2. All samples had NSTI values between 0.03 and 0.12, indicating that these OTUs made up the genomic reference database used to produce precise functional predictions. The projected pathways, which comprised a metabolism of carbohydrates, cofactors, vitamins, amino acids, terpenoids, polyketides, lipids, various secondary metabolites, and Class 2 nucleotides, were considerably enriched in the Class 1 metabolic category (Figure 7). The main functions were mainly related to metabolism, including the transport and metabolism of amino acids, carbohydrates, and inorganic ions, transcription, energy production, conversion, and biogenesis of cell membranes. The functions of the microorganisms in the Dongcai from the three regions are essentially the same.

## 4. Discussion

In this study, the HTS approach was used to examine the microbial communities of Dongcai from three different regions in eastern, central, and western China. The bacterial communities in the fermented Dongcai from the three regions varied. *Firmicutes*, *Proteobacteria*, *Bacteroidetes*, *Actinobacteria,* and other bacteria (1% abundance of species in all samples) were present in all samples. This observation is consistent with previous studies on Dongcai in SC and TJ [14]. In general, *Firmicutes* and *Proteobacteria* make up the majority of the bacteria in kimchi [15]. However, in our study, the levels of cyanobacteria in the TJ samples were very low, which is inconsistent with previous studies on TJ Dongcai [14]. Additionally, compared to the other samples, the SC samples included extremely high amounts of *Actinobacteria actinomycetes*, which may be related to their unique raw materials and preparation.

Two main groups of bacteria are considered important for vegetable fermentation: lactic acid bacteria (LAB) and halophilic bacteria (HAB) [16,17]. In our study, LAB and HAB were detected in the Dongcai from all three regions but at different relative levels. As one of the most important microbial probiotic flora, LAB has a significant influence on the physical and chemical indicators (nitrite, PH, total acid, etc.) and flavor of pickled, fermented vegetable products and is also highly beneficial [14]. In the present study, the LAB we observed in Dongcai were mainly *Weissella* and *Lactobacillus*, both dominant genera in the TJ sample. This is consistent with the findings reported in previous studies on TJ Dongcai [14]. *Lactobacillus*-related genera and *Weissella* may be more important utilizers of glucose and fructose through subsequent glycolysis, leading to the production of pyruvate, the central compound in the production of organic acids, ethanol, and esters [18,19]. Additionally, *Weissella* increases the flavor of pickled foods while simultaneously ensuring their safety by creating acid and alcohol, which prevent the growth of dangerous germs [20,21]. Medium salinity bacteria are widely used in the production of high-salt foods (fermented fish, fermented meat, soybean paste, pickles, etc.) to promote the formation of flavor substances in foods [22]. Therefore, salinophilic bacteria are also among the important microorganisms in Dongcai fermentation. We detected eight salinophilic bacteria, *Halomonas, Pseudomonas*, *Vibrio*, *Bacillus*, *Chromohalobacter*, *Salinicola*, *Gracilibacillus,* and *Virgibacillus*, in the Dongcai from the three regions. The salt content of Dongcai is high, around 10–13%, and many microorganisms cannot grow in such a high-salt environment, but it is very suitable for the growth of moderately salinophilic bacteria, which is the main reason for the presence of a large number of such bacteria in long-fermented Dongcai. In addition, the mustard used for processing is not washed, so it can be assumed that the moderately *salinophilic* bacteria in the samples may have come from raw mustard or salt. However, *salinophilic Marinococcu* and *Salinicoccus* were not detected in these samples as they have been in SC Dongcai [7], which may be related to the specificity of fermentation time, ingredients, and processing method.

Flavor compounds are a key element in the consumer acceptance and product identification of Dongcai. The formation of kimchi flavor is highly correlated with microbial fermentation, which improves the aroma and flavor attributes of kimchi through the production of aromatic substances, including organic acids, amino acids, and volatile compounds [23]. Amino acids are major contributors to the subtle flavor of fermented vegetables, and several have been shown to produce fresh or sweet flavors and may contribute to the intense freshness and sweet aftertaste of the squash [24,25]. According to our findings, Dongcai from SC exhibited the highest levels of amino nitrogen, with sample DFY11-1g reaching a level of 0.02910 ± 001 mg/mL. The high content of fresh flavor substances in the Dongcai from SC may be the primary factor contributing to their fresh flavor. In addition, total acid is also a major key to determining the flavor and quality of fermented Dongcai. We found that Dongcai from SC possessed a higher acidity level compared to the other two regions. This may be due to the fact that Dongcai produces more d of organic acids during fermentation by breaking down carbohydrates, the main component of Sichuan mustard. The higher levels of acidity and amino acid content allow Dongcai from SC to produce a unique flavor.

Further analysis of the microbial function using PICRUSt2, we found that the core functions of bacteria in Dongcai are protein metabolism and carbohydrate metabolism, which is in agreement with a previous report by Yao et al. They found that microorganisms in SC Dongcai were mainly involved in amino acid metabolism and degradation activities [26]. We found that *Weissella*, *Pantoea*, *Klebsiella*, *Lactobacillus*, *Enterobacter*, *Idiomarina* and *Pseudomonas*, *Carnimonas* and *Staphylococcus*, *Corynebacterium* and *Sphingomonas*, *Burkholderia-Caballeronia-Paraburkholderia,* and *Rhodococcus* are the eleven genera of bacteria that dominate carbohydrate metabolism in salt, acid, amino nitrogen, and sugar. This differs from the findings of previous studies that show that fungi play a major role in carbohydrate metabolism. They can produce a variety of enzymes that degrade large carbohydrates into monosaccharides, which can subsequently be metabolized by bacteria [8]. The predicted bacterial functions were also significantly enriched in the metabolism of cofactors and vitamins, terpenoids and polyketides, lipids, glycans, and nucleotides, which may explain the variety of aromas not present in Dongcai. The complex interactions between these microorganisms create the unique flavor of Dongcai, which is another source of Dongcai flavor in addition to the raw ingredients.

This study shows that the most abundant genera were Lactobacillus, Weissella, Virgibacillus, Enterobacter, Klebsiella, Pseudomonas, and Pantoea in the TJ samples, and Burkholderia- Caballeronia-Paraburkholderia, Rhodococcus, Sphingomonas, Ralstonia, and Chromohalobacter in the SC samples, while only Carnimonas was most enriched in the GZ samples. A study shows that fermented vegetables from different regions have regional similarities in bacterial communities while also having different abundances of dominant species, indicating potential differences in the fermentation production process [27]. These differences can be attributed to a number of factors. According to previous studies, the types and numbers of microorganisms in kimchi are most affected by temperature and salt concentration [28]. Salt is one of the most significant influences on the proliferation and metabolism of microorganisms during the fermentation of kimchi and the primary cause of its salty flavor [29]. Yang et al. showed that the diversity and succession of microorganisms in kimchi with different salt concentrations would have different characteristics [30]. We, therefore, hypothesize that salt concentration during curing may be a factor that plays a role in these differences in bacterial composition. The amount of salt added to SC Dongcai for pickling is generally 13–15%, while the amount added to TJ Dongcai is generally 7–12%, and the amount added to GZ Dongcai is not clearly reported at present [14,31]. It has been shown that bacterial abundance tends to increase in the early stages of fermentation in TJ Dongcai compared to SC Dongcai, probably due to the excessive salt addition (13%) compared to 7% in TJ Dongcai, which inhibits most bacterial growth. Lee et al. found that salt concentration was the main determinant of bacterial composition during kimchi fermentation and that Lactobacillus and Weissella thrived at high salt concentrations [32]. This may be because different salt concentrations result in different osmotic pressures, water activity, and strain structure, as well as rates of microbial metabolism and exchange of substances, and because each bacterium has a different tolerance to salt concentration, which directly or indirectly affects the quality and flavor of the kimchi [33]. We, therefore, speculate that the different amounts of salt added to the Dongcai cured in these three regions affect the levels of bacteria such as HAB and LAB, resulting in the different flavors of these three types of Dongcai. In addition, the temperature is also an important influencing factor because the fermentation of Dongcai is affected by the climatic conditions of different regions. In our study, Tianjin in northeastern China has a dry climate with a year-round temperature of 12–15 °C; Nanchong in Sichuan, located inland in southwestern China, has a humid climate with a year-round temperature of 16–23 °C; and Guangzhou in southern China has a humid climate with a year-round temperature of 20–28 °C. Thus, the climatic temperature differences between the three regions are very pronounced. Once fermentation begins, the number of bacteria starts to increase gradually, and the growth rate accelerates as the temperature rises. Lactobacillus is an important bacterium in the fermentation process of kimchi, with the fastest growth rate and highest acid production rate in the range of 26–30 °C [34], in particular, Weissella and Lactococcus [35]. Therefore, it is important to regulate the fermentation temperature in order to encourage the growth and multiplication of lactic acid bacteria while inhibiting the growth of dangerous microorganisms. The microbial composition and flavor in kimchi fermentation systems are related to a variety of factors, especially the raw materials. According to earlier research, the initial ecological niche of the microorganisms in the raw material dictates the final microbial community, which affects the product’s quality by generating the dominating flora during the ensuing fermentation process [36]. In particular, vegetables carry lactic acid bacteria, which are the main source of microorganisms in the fermentation process of kimchi, such as Leuconostoc, Weissella, Lactobacillus, and Lactococcus [23]. In our research, SC Dongcai mainly uses mustard, while TJ and GZ Dongcai mainly uses Chinese cabbage. The different microorganisms carried by mustard and Chinese cabbage may also be one of the reasons for the differences in the abundance of bacterial genera in fermented Dongcai from the three regions. The most distinctive feature of TJ Dongcai is the addition of garam masala, which contains about 15% garlic [37]. The study shows that the amount of LAB in kimchi increases significantly when it contains large amounts of garlic. This may be because the growth of several molds and cocci can be inhibited by allicin and ajoene, the main antibacterial components of garlic, while effectively promoting the growth of Lactobacillus and Weissella [38]. We, therefore, believe that this may be one of the reasons for the much higher amounts of Lactobacillus and Weissella in the TJ samples compared to those from other areas.

However, the traditional microbial distribution of Dongcai has a positive impact on the flavor and may, at the same time, contain harmful microorganisms. Many Gram-negative bacteria are pathogenic or spoilage bacteria that can pose a serious threat to product quality [39]. Our study did not detect several species of common pathogenic bacteria in the three regional Dongcai, namely *Bacillus* spp., *Salmonella* spp., *Staphylococcus aureus,* and *Shigella* spp. This indicates that all three Dongcai have a high safety profile.

## 5. Conclusions

Our research has revealed the microbial diversity of Dongcai from different regions of China. In Dongcai samples from TJ, SC, and GZ, we discovered varied bacterial abundances but comparable bacterial community diversity and coverage. Through HTS, we identified salinophilic bacteria (*Salmonella* and *Vibrio*, among others) and LAB (*Lactobacillus*-related genera and *Weissella*, among others) as the dominant strains in Dongcai. PICRUSt2 predicted the metabolic enrichment of amino acids, carbohydrates, polyketides, and lipids. Furthermore, we showed that complex interactions between microbes occur and that factors such as raw materials, geography, and processing techniques have an impact on microbial diversity and composition.

## Figures and Tables

**Figure 1 foods-12-01753-f001:**
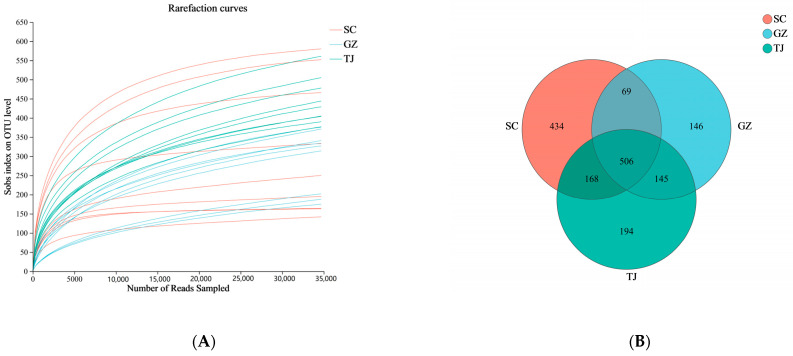
(**A**) Number of species observed in samples from different areas; (**B**) Venn diagram of the four regions based on OTUs. SC: Sichuan samples; GZ: Guangzhou samples; TJ: Tianjin samples.

**Figure 2 foods-12-01753-f002:**
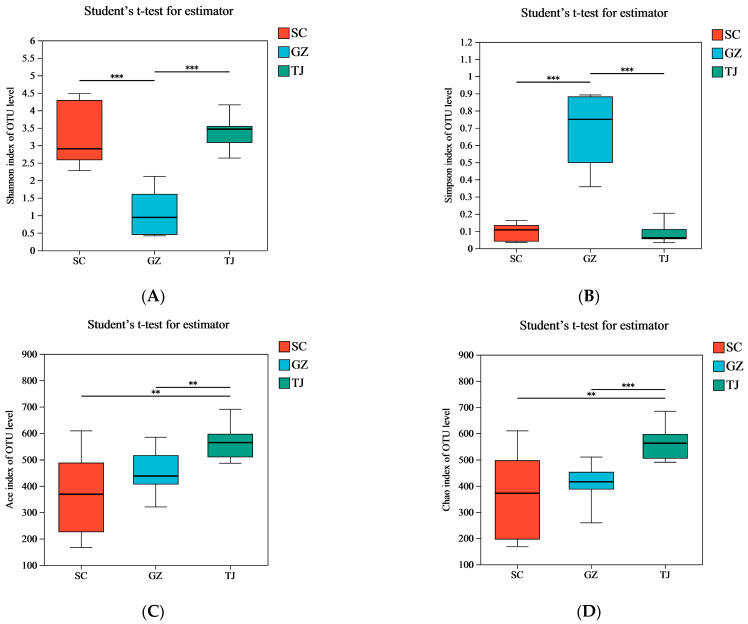
Alpha-analytical indicators of microorganisms in Dongcai from the three regions. The homogeneity and community cover of bacteria in Dongcai were assessed using the Shannon index (**A**) and Simpson index (**B**), while the richness of the microbiota in Dongcai was assessed using the ACE index (**C**) and Chao1 index (**D**). SC: Sichuan samples; GZ: Guangzhou samples; TJ: Tianjin samples. **: *p* < 0.01; ***: *p* < 0.001.

**Figure 3 foods-12-01753-f003:**
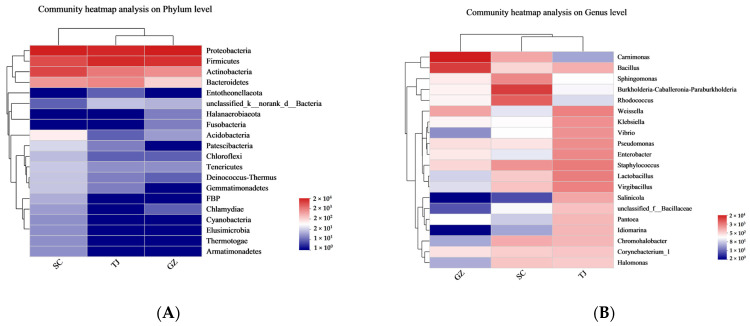
(**A**) Cluster heat map of bacterial phyla in Dongcai samples from the three regions; (**B**) Cluster heat map of bacterial genera in Dongcai from the three regions. SC: Sichuan samples; GZ: Guangzhou samples; TJ: Tianjin samples.

**Figure 4 foods-12-01753-f004:**
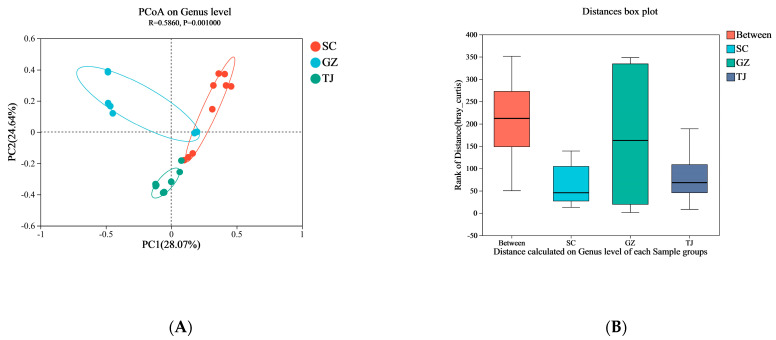
(**A**) Bray–Curtis distance PCA plot for microbial Genus levels; (**B**) Similarity analysis based on distances calculated at the genus level. SC: Sichuan samples; GZ: Guangzhou samples; TJ: Tianjin samples.

**Figure 5 foods-12-01753-f005:**
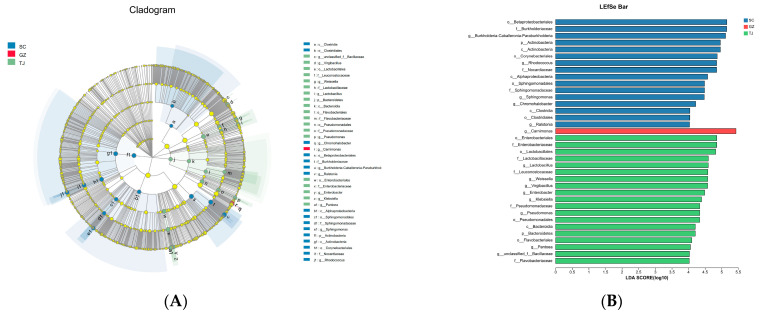
Different bacterial community architectures in Dongcai were examined using LDA LEfSe techniques: (**A**) Differences in statistical and biological consistency between the two sample groups are represented taxonomically, and significant differences with an LDA threshold of 4.0 are represented by colored branches; (**B**) LDA score histogram for hetero-rich genera. SC: Sichuan samples; GZ: Guangzhou samples; TJ: Tianjin samples.

**Figure 6 foods-12-01753-f006:**
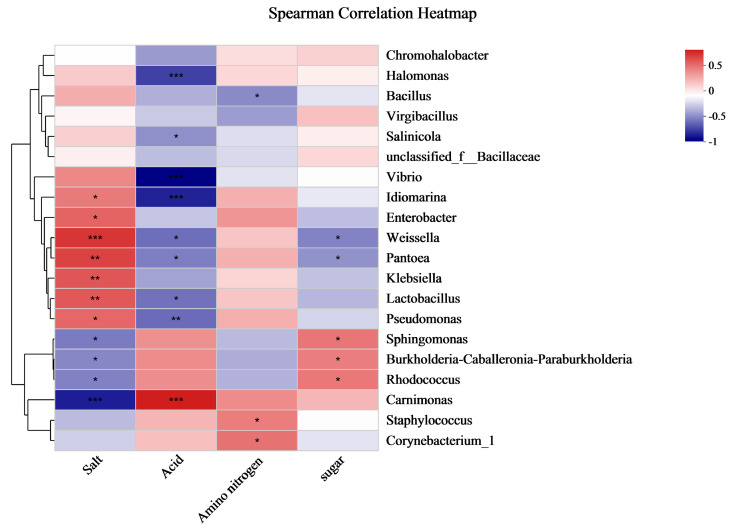
Spearman correlation heat map of microbial community structure and biochemical indicators of Dongcai. SC: Sichuan samples; GZ: Guangzhou samples; TJ: Tianjin samples. The right-hand color card of the heat map shows the color partitioning of the different R values, with red representing positive correlations, 0 < R < 0.5, and blue representing negative correlations, −1 < R < 0. *: *p* < 0.05; **: *p* < 0.01; ***: *p* < 0.001.

**Figure 7 foods-12-01753-f007:**
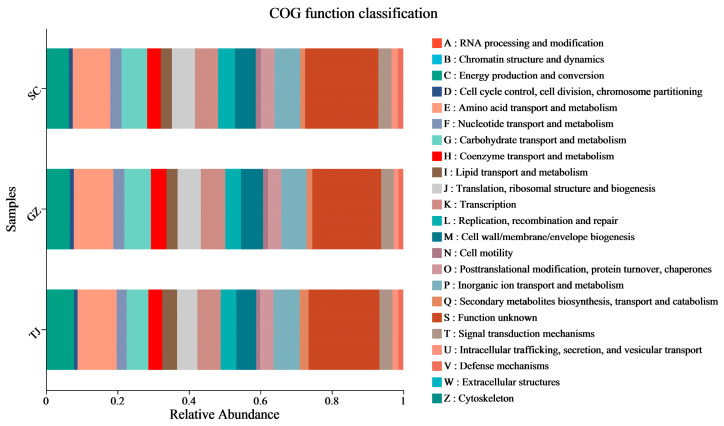
Predicted functional profile of PICRUSt2 in microorganisms. SC: Sichuan samples; GZ: Guangzhou samples; TJ: Tianjin samples.

**Table 1 foods-12-01753-t001:** Situation of sample classification.

Sample Origin	Code	Sample Origin	Code	Sample Origin	Code
	DFY11_1		DBT22_1		DQY32_1
	DFY11_2		DBT22_2		DQY32_2
	DFY11_3		DBT22_3		DQY32_3
	DTG12_1		DLY22_1		DFN31_1
Sichuan	DTG12_2	Guangzhou	DLY22_2	Tianjin	DFN31_2
	DTG12_3		DLY22_3		DFN31_3
	DZY13_1		DXJ23_1		DCC33_1
	DZY13_2		DXJ23_2		DCC33_2
	DZY13_3		DXJ23_3		DCC33_3

**Table 2 foods-12-01753-t002:** Biochemical data of Dongcai from the three regions.

	Sample ID	Salt (mg/mL)	Acid (mg/mL)	Amino Nitrogen (mg/mL)	Sugar (mg/mL)
	DFY11_1	0.1848 ± 0.004	0.0558 ± 0.002	0.0291 ± 0.001	20.78 ± 0.67
SC	DTG12_1	0.1741 ± 0.003	0.0628 ± 0.001	0.0271 ± 0.001	20.33 ± 0.51
	DZY13_1	0.1569 ± 0.009	0.0471 ± 0.002	0.0193 ± 0.006	20.5 ± 0.11
	DBT22_1	0.1885 ± 0.003	0.0537 ± 0.001	0.0221 ± 0.008	21.01 ± 0.62
GZ	DLY22_1	0.2204 ± 0.004	0.0537 ± 0.001	0.0222 ± 0.008	21.14 ± 0.36
	DXJ23_1	0.2427 ± 0.007	0.0422 ± 0.002	0.0162 ± 0.006	12.56 ± 0.96
	DFN31_1	0.2461 ± 0.001	0.0535 ± 0.003	0.0281 ± 0.001	20.53 ± 0.73
TJ	DQY32_1	0.4532 ± 0.004	0.0368 ± 0.005	0.0132 ± 0.004	20.68 ± 0.46
	DCC33_1	0.5138 ± 0.01	0.0231 ± 0.001	0.005 ± 0.002	20.42 ± 0.87

SC: Sichuan samples; GZ: Guangzhou samples; TJ: Tianjin samples.

## Data Availability

Data is contained within the article.

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
