# Peer review of "Characterization of Functional Microorganisms in Representative Traditional Fermented Dongcai from Different Regions of China"

_foods, 2023, doi:10.3390/foods12091753_

Round 1
Reviewer 1 Report
The study is interesting since, it employed the high-throughput sequencing method to examine the microbial biodiversity of Dongcai from three distinct regions in China. This study's findings add to existing knowledge of the Dongcai region's microbial diversity and point the way for more effective utilization of useful microbes in the future. There is regional variation in the prevalence of the primary species of fermented Dongcai. Carbohydrates, amino acids, polyketides, lipids, and other secondary metabolites are predicted to be readily accessible for biosynthesis.On the other hand, Salts, acids, amino nitrogen, and sugar were found to have a substantial positive link with the prevalence of HAB and LAB, and it was speculated that this would account for the varying flavors associated with Dongcai.
The study is interesting, however, the following minor corrections are necessary.
1. The quality of figure 1,2 3, 4 and 5 must be improved. Its hard to see the data presented on these figures
2. Under section 3.2. The Correlation analysis of microbial communities with acids was not mentioned. This is very important observation that must be discussed as well.
3 Under the discussion section, line 230-232 must be removed. This is not part of the discussion
4. Grammar and spell checks are necessary. Just to mention a few examples;
line:300. There must be a space between Dongcai and Dynamic changes.
Line 380; there must be a space between dominant and Dynamic changes.
Line 385; there must be a space between garlic and Development.
Line 386; the open bracket is missing
Line 397; there must be a space between effect and Effect of fermentation
Line 413; there muast be a space between spoilage and Isolation
5. Several discrepancies/inconsistencies in referring and citing have been observed. For example, when citing their sources, authors sometimes provide the year alongside the reference number, while other times they include the title alongside the reference number, etc.
Author Response
Dear Editor and reviewer,
Thank you very much for giving us this opportunity to revise our manuscript. Your comments and suggestions on our manuscript have been encouraging and helpful. The following pages provide a detailed point-by-point response to your comments. Note that the reviewer’ comments are presented in Italics, and our responses are in Roman and blue font. In addition, we addressed all these major points and other issues carefully and revised the manuscript accordingly, and the manuscript be marked up using the “Track Changes” function. Please let me know if you have any further questions.
In response to Reviewer 1's suggestion, we have made the following modifications and responded:
- The quality of figure 1,2 3, 4 and 5 must be improved. Its hard to see the data presented on these figures
Reply: Thanks to the reviewer for reviewing this paper and for the question, and we have reworked the images.
- Under section 3.2. The Correlation analysis of microbial communities with acids was not mentioned. This is very important observation that must be discussed as well.
Reply: Thank you for reviewing this with such care . We have made the relevant changes to section 3.2 and the discussion.
- Under the discussion section, line 230-232 must be removed. This is not part of the discussion
Reply: Thanks to the reviewer's question. This was an oversight on our part and we have removed it.
- Grammar and spell checks are necessary.
Reply: Thanks to the reviewer's question. These were our mistakes and we have corrected the wording.
- Several discrepancies/inconsistencies in referring and citing have been observed. For example, when citing their sources, authors sometimes provide the year alongside the reference number, while other times they include the title alongside the reference number, etc.
Reply: Thanks to the reviewer's suggestion. These were our mistakes and we have revised it.
Thank you again for your advice, all of your suggestions are very important and they will guide me in my thesis writing and research work!
Please see the attachment.

Reviewer 2 Report
I reviewed the “Characterization of Functional Microorganisms in Representative Traditional Fermented Dongcai from Different Regions of China” entitled manuscript.
In my opinion, since the article mentions a local food product, it will not have a widespread effect.
Reviewer 3 Report
The names of the microorganisms are not italicized throughout the paper.
Three samples per region are few. The authors work three parallels out of one sample, it would be more accurate to analyze more samples per region for differences between regions.
Table 1 - What is the meaning of the new designations of the samples with the designations DFY, DBT, DQY, which are hardly used in the paper, but the designations SC, GZ, TJ are used?
In Figure 1, it would be appropriate to use the same color for each sample in all images. NPR. in Figure 1a SC is red and in Figure 1b SC is blue...
The text on the images needs to be larger, e.g. in Figure 3 it is invisible, i.e. illegible.
Table 2 is questionable. Where do these results come from? Where are the methods for determining these metabolites described in Materials and Methods?
The discussion is quite extensive and tedious.
Round 2
Reviewer 2 Report .
Reviewer 3 Report
It's nice that you took the comments into account and adjusted the colors on the pictures. But you failed to match the colors in the 4B and 5B images.